# A Secure and Smart Home Automation System with Speech Recognition and Power Measurement Capabilities [note 1]

**DOI:** 10.3390/s23135784

**Published:** 2023-06-21

**Authors:** Chandra Irugalbandara, Abdul Salam Naseem, Sasmitha Perera, Sithamparanathan Kiruthikan, Velmanickam Logeeshan

**Affiliations:** 1Department of Electrical Engineering, University of Moratuwa, Moratuwa 10400, Sri Lanka; chandra.legendary@gmail.com (C.I.); nazeemthebeta@gmail.com (A.S.N.); sasmithahasanthip@gmail.com (S.P.); kiruthikan011@gmail.com (S.K.); 2BCS Technology International Pty Ltd., Colombo 00700, Sri Lanka

**Keywords:** home automation systems, speech recognition, natural language understanding, smart plug socket

## Abstract

The advancement in the internet of things (IoT) technologies has made it possible to control and monitor electronic devices at home with just the touch of a button. This has made people lead much more comfortable lifestyles. Elderly people and those with disabilities have especially benefited from voice-assisted home automation systems that allow them to control their devices with simple voice commands. However, the widespread use of cloud-based services in these systems, such as those offered by Google and Amazon, has made them vulnerable to cyber-attacks. To ensure the proper functioning of these systems, a stable internet connection and a secure environment free from cyber-attacks are required. However, the quality of the internet is often low in developing countries, which makes it difficult to access the services these systems offer. Additionally, the lack of localization in voice assistants prevents people from using voice-assisted home automation systems in these countries. To address these challenges, this research proposes an offline home automation system. Since the internet and cloud services are not required for an offline system, it can perform its essential functions, while ensuring protection against cyber-attacks and can provide quick responses. It offers additional features, such as power usage tracking and the optimization of linked devices.

## 1. Introduction

The popularity of home automation systems is increasing in a fast developing world, offering comfort, convenience, and safety to users. These systems, especially those that are voice-activated, have particularly helped elderly people and those with disabilities [1]. These systems are developed with a central controller, that controls appliances, such as power outlets, temperature sensors, lights, security systems, and emergency systems. A major advantage to the user is that the connected devices can be managed and controlled remotely using various devices, such as smartphones, laptops, tablets, desktops, and even voice commands in the latest home automation systems.

The widespread availability of Wi-Fi-enabled devices has exponentially increased the adoption of smart home systems [1]. These systems offer benefits, such as energy savings, ease of use, time conservation, and a better quality of life. As the internet of things continues to advance, home automation system developers are now utilizing cloud-based systems.

The popularity of home automation systems with voice assistance has also risen among consumers. Devices, such as Amazon Echo, Google Home, and Apple HomePod, have become integral to the smart home experience. Many smart appliance manufacturers have incorporated one or more of these voice assistants to increase the sales of their products.

Speech recognition is a crucial component in home automation systems. In the existing automation systems, voice recognition services are performed by third-party cloud services, such as Wit.ai, IBM Watson, Google Cloud Speech, Microsoft Cognitive Services, etc. However, if the internet connection is unstable, the dependency on cloud services can lead to the failure of voice recognition features.

Additionally, these devices must be always connected to the internet to maintain the connection to the cloud, usually through the home Wi-Fi. With increasing cyber-security threats, Wi-Fi connections are becoming increasingly vulnerable, and having a system that can control the household devices attached to it makes it much more dangerous for users. There have been allegations that companies, such as Google and Amazon, have acquired private information from users through their home automation systems, which raises concerns about privacy violations [2]. This results in a trade-off between user privacy and convenience that should not be ignored.

Furthermore, while well-known home automation systems are popular and reliable in some areas of the world, they are highly unreliable in developing nations due to poor internet connectivity. In addition, the lack of localization in voice assistants also restricts the use of home automation systems in developing countries, which makes the home automation industry restricted to specific regions of the world.

The aforementioned limitations of home automation systems are due to the reliance on cloud-based services [3]. This creates a need for an offline-based voice-assisted home automation system. Several studies have proposed different methods to implement an offline-based, voice-assisted, home automation system that does not depend on cloud-based services. However, there are several limitations in their implementation, which are discussed in the literature review section. This research proposes an offline voice-assisted home automation system that mitigates these limitations and has significant improvement in performance. In addition, it requires less memory and lower computational requirements compared to the existing methods. The developed system includes a compatible smart plug in addition to the basic smart home functionality, such as voice assistance, relay control, and energy consumption tracking with improved privacy.

This paper is organized as follows. Section 2 discusses the literature review. Section 3 presents the architecture of the system. Section 4 describes the implementation of the proposed approach. Section 5 presents the results of the evaluation carried out. Finally, Section 6 concludes the outcome of the research.

## 2. Literature Review

Researchers have proposed various methods to implement offline voice-assisted home automation systems. Errobidart et al. proposed an offline demotic system with voice commands, and is shown in Figure 1 [4]. The system consists of an EasyVR speech recognition module and an Arduino Mega. Since it lacks natural language processing (NLP), a pre-defined set of commands must be assigned to each task. In addition, the cost of the EasyVR shield and the Arduino Mega raises the total cost of the device to a level higher than existing home automation systems on the market [5]. The system was developed with two communication protocols. However, it can only maintain a proper connection up to 50 cm.

G. et al. [6] proposed a method that employs the hidden Markov model toolkit (HTK) to convert speech into text. The overview of the system is shown in Figure 2. It uses a GSM module to transfer signals between the hub and smart plug via SMS. However, the HTK has limitations in terms of shorter time intervals that impacts performance. Natural language processing (NLP) is not utilized in this system either. In addition, the hub does not have an inbuilt microphone.

Elsokah et al., proposed a next-generation home automation system that is based on voice recognition and that uses an Easy VR 2.0 shield in combination with an Arduino microcontroller [7]. The overview of the system is shown in Figure 3. The system communicates between the hub and smart plug through a Wi-Fi module and has the added advantage of incorporating environmental inputs, such as humidity and temperature. However, the number of commands that can be executed is limited due to the use of the Easy VR shield.

Rani et al. proposed a system that utilizes natural language processing (NLP) and uses a mobile phone for voice input and processing, as illustrated in Figure 4 [8]. The commands are then sent to the Arduino, that acts as a controller in a smart plug through Wi-Fi. However, a significant drawback of this system is its reliance on a mobile phone.

Our proposed offline home automation system delivers a significant advancement in the field of smart homes. One of the primary advancements of our system is the significant performance improvements over existing home automation systems. Our system features faster response times, higher accuracy, and efficient utilization of resources. This enables users to control their smart homes quickly, without experiencing frustrating delays or inaccurate responses.

Another key advantage of our system is the reduction in memory and computational requirements. This results in lower hardware requirements and reduced power consumption than the existing systems, which makes it more accessible and affordable to a wider range of users.

Overall, our offline home automation system represents a significant step forward in the field of smart homes. Due to the improved performance and reduced computational requirements. Our system provides valuable benefits to the users who are looking to simplify and streamline their home automation tasks.

## 3. System Architecture

The proposed “HomeIO” is a low-cost, offline, and versatile home automation system with built-in speech recognition and intention detection. Its goal is to simplify the system and reduce its cost. This will eliminate the need for high-performance cloud computing which in turn will reduce the privacy and cyber security risks. The system is also flexible, allowing new appliances from other manufacturers to be easily added to the network for safe and secure operation. Additionally, the system includes smart plug sockets with improved distance connectivity and energy measurement capabilities. Figure 5 shows the system architecture consisting of a smart hub and smart plug socket.

This section is divided into three subsections. The first subsection describes the smart hub component. The second subsection describes the smart plug socket component, and the third subsection describes the communication protocols and mediums used for connectivity.

### 3.1. Smart Hub

In a home automation system, the smart hub serves as the central control point for intercommunication among components. There can be one or more smart home hubs or none at all. Typically, smart hubs rely on cloud processing to handle the demands of voice assistants, which can result in inoperability if cloud services are unavailable. To overcome this, HomeIO’s smart hub features an on-device speech-based user interface, that allows users to communicate with the system without the need for third-party services.

Figure 6 shows an overview of the speech recognition system used in HomeIO. The system consists of four components: a voice activity detection (VAD) model, a wake word detection model, a speech-to-text model, and a natural language understanding model. The VAD model has an algorithm that constantly monitors the surroundings for speech signals and uses VADs to identify speech segments with lower energy consumption. Upon detecting a voiced segment, the wake word detection model is triggered. If the wake word is detected, the speech-to-text model converts the audio signals into sentences. The natural language understanding engine determines the scenario, intent, and entity from the sentence. This is passed on to the control system to make a decision.

#### 3.1.1. Voice Activity Detection (VAD)

Voice activity detection (VAD) is an essential component of many speech signal processing programs that separates audio streams into periods of speech activity and periods of silence. It operates continuously when the device is on. Therefore, its algorithm should have a low energy consumption. VAD recognizes when a person is speaking to the device and helps determine when the person has stopped speaking. The VAD algorithm used in this proposed home automation system is based on the open-source Google WebRTC voice activity detector, written in C language for real-time web communications [9]. The system uses Gaussian mixture models (GMMs) to distinguish between voiced and unvoiced speech segments. The VAD only supports 16-bit mono PCM audio with several preset sample rates and frame intervals.

#### 3.1.2. Automatic Speech Recognition (ASR)

Since the speech recognition in HomeIO takes place on the device, the models used must be optimized for small sizes and low computational demands. In order to achieve this, the speech recognition system is divided into two main components: the speech-to-text model and the natural language understanding (NLU) model.

The goal of the speech-to-text (STT) model is to analyse an audio signal, break it down, digitize it into a machine-readable format, and produce the most appropriate text representation. In order to achieve this, the speech-to-text systems rely on two types of models: acoustic models and language models. Our implementation of STT utilizes a simple convolutional neural network (CNN), as shown in Figure 7. The model predicts the letters pronounced by the user, and once there is a silence between words, the predicted letters are passed through a connectionist temporal classification (CTC) beam search decoder with lexicon constraints and a language model to obtain the best estimate of the sentence. the model was trained on Mozilla’s common voice English dataset that contains 2886 h of audio data labelled by 79,398 different voices. For the language model, a pre-trained KenLM model is used.

The input sentence must be translated into a machine-readable format for the smart hub to understand and execute the instruction. In the past, simple if-else statements were used to check for specific terms in the text, but these methods cannot capture key aspects, such as time, location, and intention. With advancements in NLP and ML, it is now possible to extract meaningful information from a sentence. Our system uses a pre-trained bidirectional encoder representations from transformers (BERT) model. The overview of this model is shown in Figure 8. It extracts the mask from the input sentence, that is then sent to an artificial neural network (ANN), that determines the scenario, intention, and entity from the input sentence [10,11]. An example of this would be the sentence “Turn off the light in the kitchen”, where the scenario is “IoT”, the intention is “Turn OFF”, and the entities are “Kitchen: location” and “lights: device”. These variables can then be used in control logic to perform the desired task. The model was trained using open-source data.

#### 3.1.3. Device Power and Security Management

The HomeIO system includes a feature to monitor the power usage of the connected devices. This is accomplished through an on-device database that tracks power consumption and temperature readings from sensors on smart devices, and provides real-time updates to the user. This information can be used to set schedules and rules for switching on and switching off of the devices. The mesh network’s security is ensured through a two-layered security channel, where a user must know the username and password for the smart hub and the decryption key for each connected device to access it.

### 3.2. Smart Plug Socket

The smart plugs are designed to be placed between the wall socket and electrical appliances, that enables the user to switch on and switch off the appliance using a smartphone or voice commands via wireless communication. Smart plugs have become a necessity in home automation systems as they help save household energy while maintaining the user’s comfort and enhancing their way of life. In comparison, using traditional power sockets requires physical interaction to switch on and switch off the appliances, and they also lack the ability to remotely monitor the appliance’s status or measure power usage.

While some smart plugs on the market have energy management capabilities, many brands only offer this feature in a more expensive pro version due to the size and cost of the energy measurement unit. Measuring energy usage is crucial in promoting a more sustainable and environmentally friendly energy use [12].

#### 3.2.1. Power Usage Tracking

According to a study by Ahmed et al. in 2015, a power node was developed to monitor the power usage in smart plugs [13]. It consisted of a voltage sensor, a current sensor, and a Zigbee microcontroller. However, this design has limitations, such as a slow sampling rate, poor communication coverage, and high cost.

The proposed smart plug design includes a feature to monitor the real-time power consumption of an electrical appliance. In the design, HLW8012, a single-phase energy monitoring integrated circuit, is used to gather data, such as RMS current, RMS voltage, and RMS active power. This IC is directly connected to an ESP32 Node MCU, that allows for fast power usage calculations. The use of an HLW8012 power sensor instead of separate current and voltage sensors makes the design more compact and cost-effective.

#### 3.2.2. Relay Operation

The smart plug uses a 5 V electro-mechanical relay to control the connected appliances. The relay is activated by a DC current, that opens or closes the switch contacts. The ESP32 Node MCU receives commands from the user, either through a voice command or a control action using the mobile app, and operates the relay contacts accordingly.

### 3.3. Connectivity

The connectivity between the hub and devices is a crucial aspect of any home automation system, as it determines its reliability. Communication protocols, such as BLE, Zigbee, Z-Wave, and Wi-Fi are commonly used by home automation systems, each with its own advantages and disadvantages [14]. Factors, such as range, bandwidth, interference resistance, and energy consumption, impact the stability of the connection. The cost-effectiveness, security, and ease of configuration are important considerations for customers [15].

When choosing a communication protocol for HomeIO, which is an offline home automation system that prioritizes security and reliability, factors, such as resistance to external attacks, sufficient bandwidth for real-time data collection from power-monitoring devices, and ease of connection, must be considered. Therefore, a Wi-Fi-based wireless mesh network is the best communication solution for HomeIO.

In order to mitigate the inherent constraints of stand-alone networking systems, such as signal loss as devices move away from the router and interference from electrical equipment, mesh networking was selected. Mesh networks can self-organize and configure dynamically, resulting in reduced installation time. Self-configuration enables dynamic workload distribution. This is useful if several nodes fail simultaneously and results in an improved fault tolerance and maintenance costs. The use of mesh networking in home automation, combined with a suitable communication protocol, creates a reliable, low-maintenance, and fault-tolerant device-to-device or hub-to-hub connection that can work well even in the presence of walls or other electronic devices that may interfere with communication.

Message queuing telemetry transport (MQTT) is a standard IoT messaging protocol developed by the Organization for the Advancement of Structured Information Standards (OASIS). The overview of the protocol is shown in Figure 9. It is a widely used messaging protocol in the IoT field because of its simplicity and security features. It has a low overhead for both coding and network traffic. It uses transport layer security (TLS) encryption and authenticates clients using an open-standard authorization (OAuth) protocol to ensure secure communication between devices [16]. In the HomeIO system, MQTT will be used as the messaging protocol due to its ease of use and compatibility with the programming languages used in the smart hub and smart devices.

## 4. Implementation

The proposed system consists of two separate devices, a smart hub and a smart plug socket that were created separately and were tested for communication. The prototype includes an offline speech recognition system, power monitoring capability, and the ability to control the smart plug through a relay. The two components are discussed in detail in the following subsections.

### 4.1. Smart Hub

#### 4.1.1. Hardware Implementation

The smart hub device consists of a Raspberry Pi 4 with 4 gigabytes of memory, a 5 V/2 A power adapter connected to its USB-C port, and a ReSpeaker 2-mic Array Pi HAT as the microphone.

#### 4.1.2. Software Implementation

The configuration of the speech recognition system is optimized for the highest accuracy possible. The voice activity detection model runs continuously while the device is on. All of the other models (wake-word detection, speech-to-text, and natural language processing) are executed in a single script. The firmware components are stored on an SD card, since the Raspberry Pi module does not have an internal memory. The microphone listener event is executed in a separate thread and updates a global circular queue. The VAD takes chunks of audio from the queue using a specific chunk size and determines if the chunk contains speech. If there are 10 consecutive chunks without speech, the process stops and the full command, including the wake-word, is transcribed to text and parsed through the natural language processing for entities, intent, and scenario. These are then sent to the controller to carry out the desired task.

### 4.2. Smart Plug Socket

For the smart plug socket, an ESP32 NodeMCU microcontroller was chosen to gather sensor data and act as the controller. The necessary codes for the relay operation and energy measurement were written using the Arduino Integrated Development Environment (IDE).

The smart plug socket is powered by a 5 V DC power supply module. To ensure extra safety, a fuse has been included in the relay control circuit design. The 5 V relay has five pins: End 1, End 2, Common (COM), Normally Closed (NC), and Normally Open (NO). End 1 and End 2 pins are used to activate the relay. These are connected to 5 V and the ground, respectively. To control a household appliance, one end should be connected to the common (COM) pin of the relay and the other end should be connected to either the Normally Open (NO) or Normally Closed (NC) pin.

The smart plug socket uses the HLW8012 breakout board for energy measurement. A schematic and a printed circuit board have been designed for the smart plug socket, where all of the modules can fit in a small space. The schematic diagram of the smart plug socket is shown in Figure 10. The smart socket plug is designed to monitor the power usage when it is not connected to the hub, and to transmit the recorded data when it is connected to the hub.

### 4.3. Connectivity

The communication between the smart hub and the smart plug socket is facilitated by Node-RED installed on the Raspberry Pi. The Raspberry Pi functions as the server, while the NodeMCU functions as a client. The MQTT protocol is employed to enable two-way communication with the MQTT broker located in the smart hub, and acts as the server that receives messages from the clients (the smart plug socket) in the network. The mac address of the NodeMCU in the smart plug socket is used as the identifier to establish connections within the Wi-Fi mesh network. To connect a new smart plug socket to the hub, a user needs to scan a QR code or enter the mac address via a mobile app made to connect devices and display the energy consumption. Then, they can assign the device a name (e.g., “kitchen light”), type (e.g., “light”), and location (e.g., “kitchen”), that allows them to control the specific smart plug socket through commands.

### 4.4. Dashboard

The system includes a dashboard created using ReactJS as the central interface for users to connect with the smart hub and smart plug socket. The interface of the dashboard is shown in Figure 11. Figure 12 shows the measured power usage data of a selected appliance. This dashboard serves as a tool for managing devices and monitoring them, and runs locally on the smart hub. It can be accessed from any device that is connected to the same Wi-Fi mesh network and provides the following user-friendly features to make the operation of the home automation system easier.

Ability to customize the connected devices to the hub;Monitor/Change the on-off status of the connected devices;Gives a graphical view of the energy consumption data.

This user-friendly dashboard gives the user the ability to easily adjust the number of connected devices. Devices can be quickly added or removed from the network, and the real-time on-off status of each device can be monitored. The user can also turn the devices on or off with a simple toggle operation, which enables remote control and reduces energy waste.

The dashboard plays a crucial role in helping users understand their energy consumption patterns by providing detailed information about the energy consumption of their home for a specified period of time. This can aid in reducing the energy waste and promote energy efficiency.

## 5. Evaluation and Results

Evaluation of the smart plug was carried out in stages, starting with individual units and then as a complete system. The accuracy of the power measurement in the smart plug ranges from 90–95%. The wireless mesh network was able to efficiently self-organize and continuously transmit data without failure within a range of 8–10 m.

The automatic speech recognition (ASR) system was installed on a Raspberry Pi 4 with 4 GB RAM. The system required 50% of the RAM and less than 40% of the CPU for simultaneous operation of all three models, leading to fast results. The ASR was able to accurately detect English language commands and execute relay operations. Results showed that HomeIO’s STT models performed better than other popular STT models in terms of word error rate (WER) and memory usage, as shown in Table 1. Compared to the existing models, our model requires significantly less memory, and has fewer parameters, without a significant drop in accuracy. Memory requirements are reduced by using quantization and pruning. Lazy loading is utilized to reduce the inference time of the neural network, which enables to reduce the computational requirements without increasing the response time. Among the existing models, Quartz Net [17] has the lowest word error rate. However, it has 18.9 Million parameters and requires significantly high memory and computational power. Compared to this model, our model has only 4.3 million parameters, which is nearly 75% less. This saves a lot of memory and computational power, which results in a lower power consumption. Even though the our model requires 180 MB of storage, it can be stored on an SD Card, which is economically feasible. The accuracy of the NLU model in a private dataset was as high as 96%.

## 6. Conclusions

The advancement in the internet of things (IoT) has revolutionized home automation systems and has made people’s lives more convenient and comfortable. Voice-assisted home automation systems have been especially beneficial for elderly people and those with disabilities. However, the dependency on cloud-based services has made these systems vulnerable to cyber-attacks, particularly in developing countries where the quality of the internet is low. To address these challenges, this research proposes an offline home automation system that operates independent of internet and cloud services. This system ensures protection against cyber-attacks, provides quick responses, and offers additional features, such as power usage tracking and the optimization of linked devices.

There are several potential areas for improvement that we can explore in the future. One potential area for improvement is the accuracy. While our system has already demonstrated impressive results, we believe that there may be room for further improvement. We are currently experimenting with different deep learning model architectures and parameter configurations to find the optimal setup for an automation system. We can also focus on enhancing the user experience of our system. This involves developing a more intuitive user interface. By improving the user experience, we can make our system more appealing to a wider range of users.

Finally, we can focus on enhancing the security of our system. Even though our system does not rely on cloud-based services, it relies on Wi-FI mesh networks to function, which has a certain level of security concern. We can focus on improving the security of our system.

By exploring these potential areas for improvement, we can ensure that our offline home automation system remains competitive and continues to provide value to our users.

## Figures and Tables

**Figure 1 sensors-23-05784-f001:**
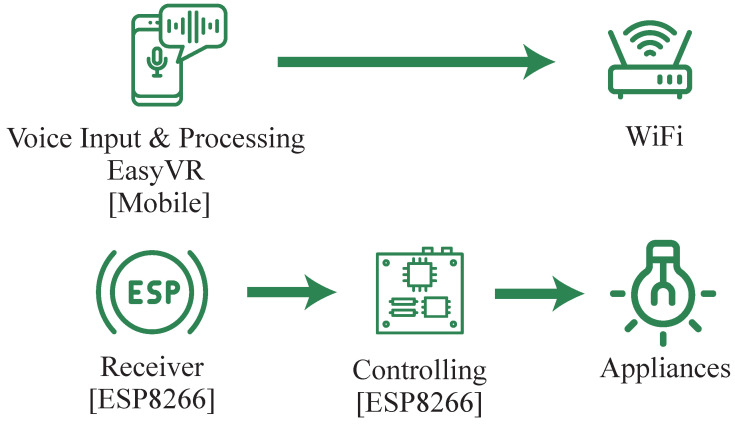
Offline demotic system using voice commands.

**Figure 2 sensors-23-05784-f002:**
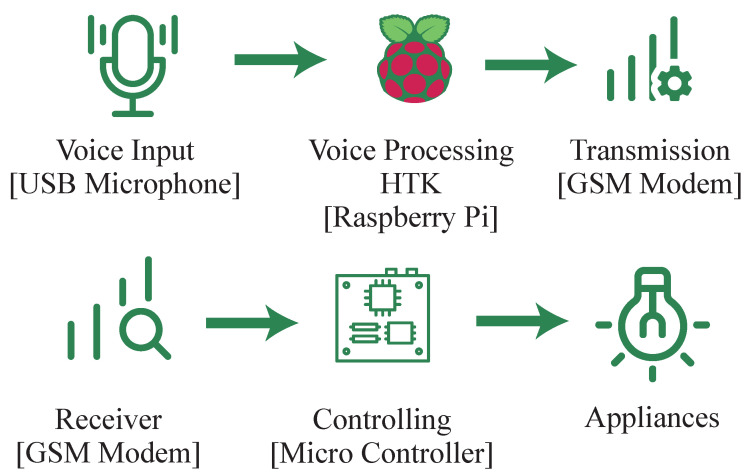
Low-cost home automation system using offline speech recognition.

**Figure 3 sensors-23-05784-f003:**
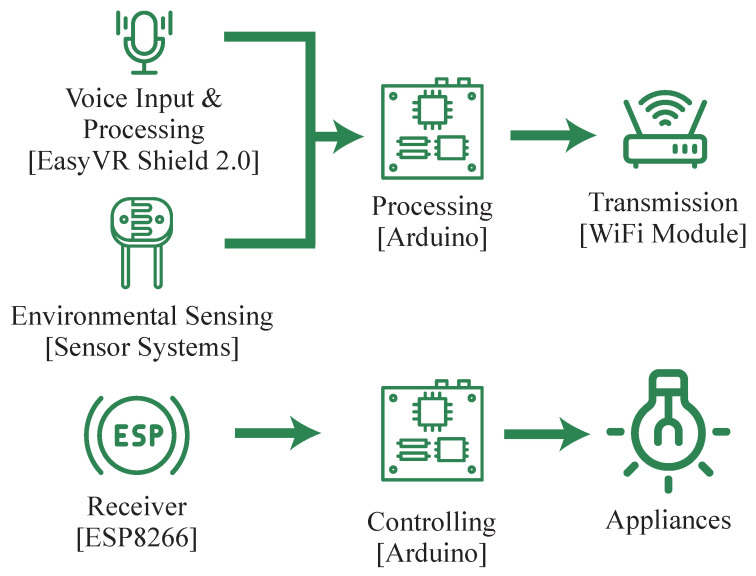
Next generation home automation system based on voice recognition.

**Figure 4 sensors-23-05784-f004:**
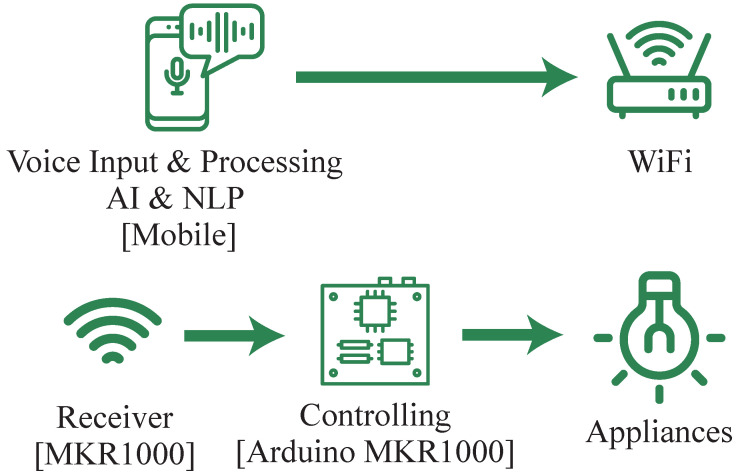
Voice-controlled home automation system using natural language processing and the internet of things.

**Figure 5 sensors-23-05784-f005:**
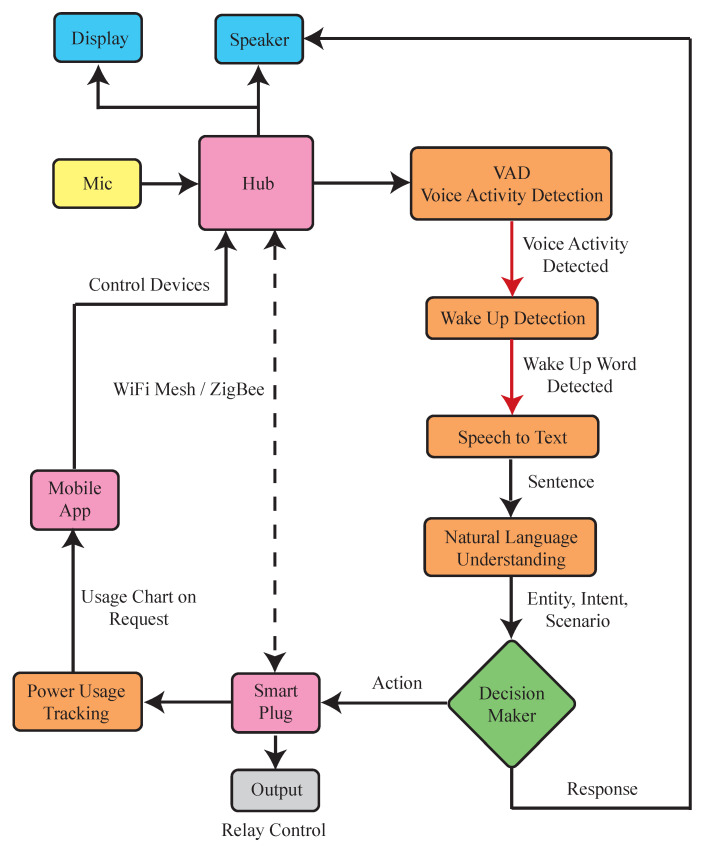
System architecture.

**Figure 6 sensors-23-05784-f006:**
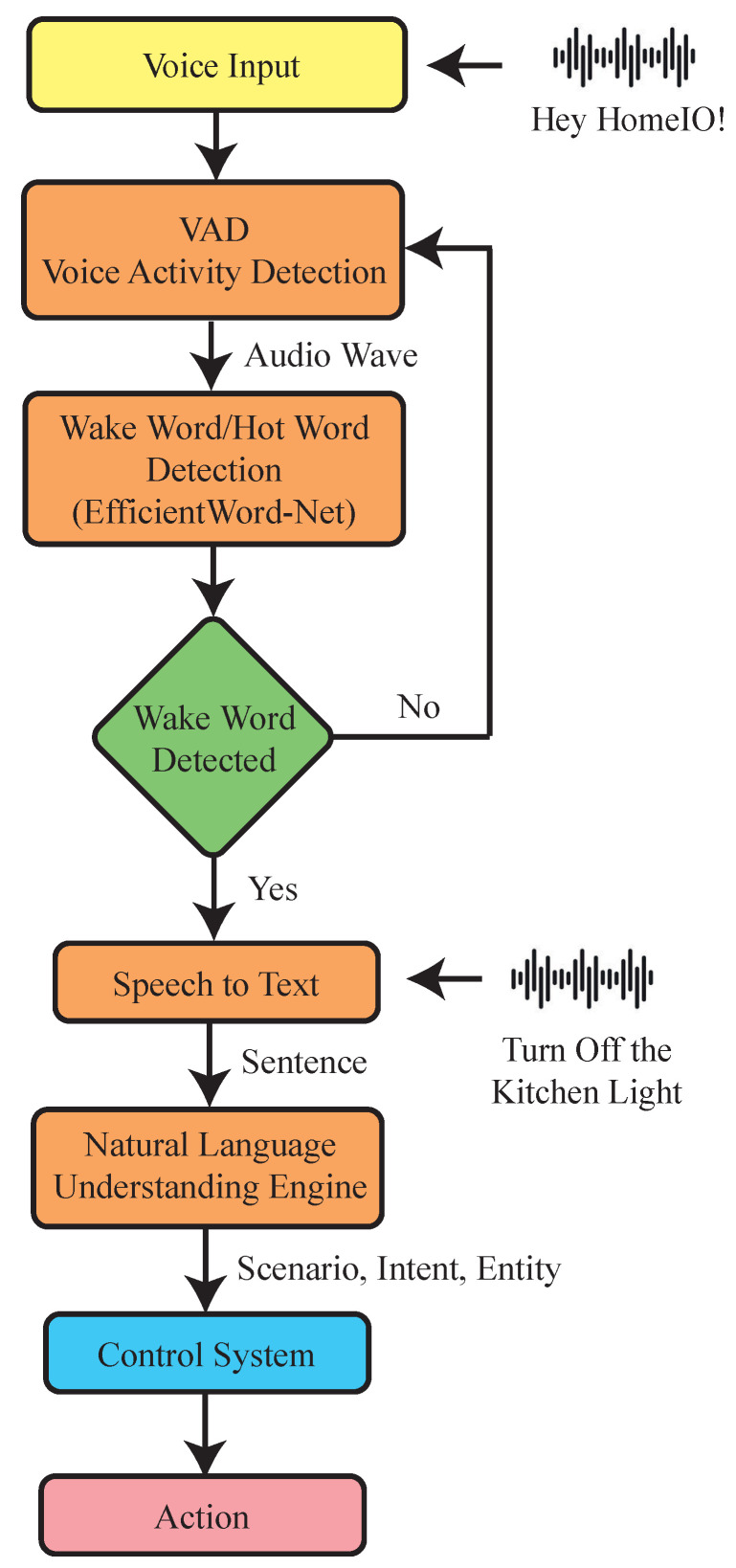
Overview of the speech recognition system.

**Figure 7 sensors-23-05784-f007:**
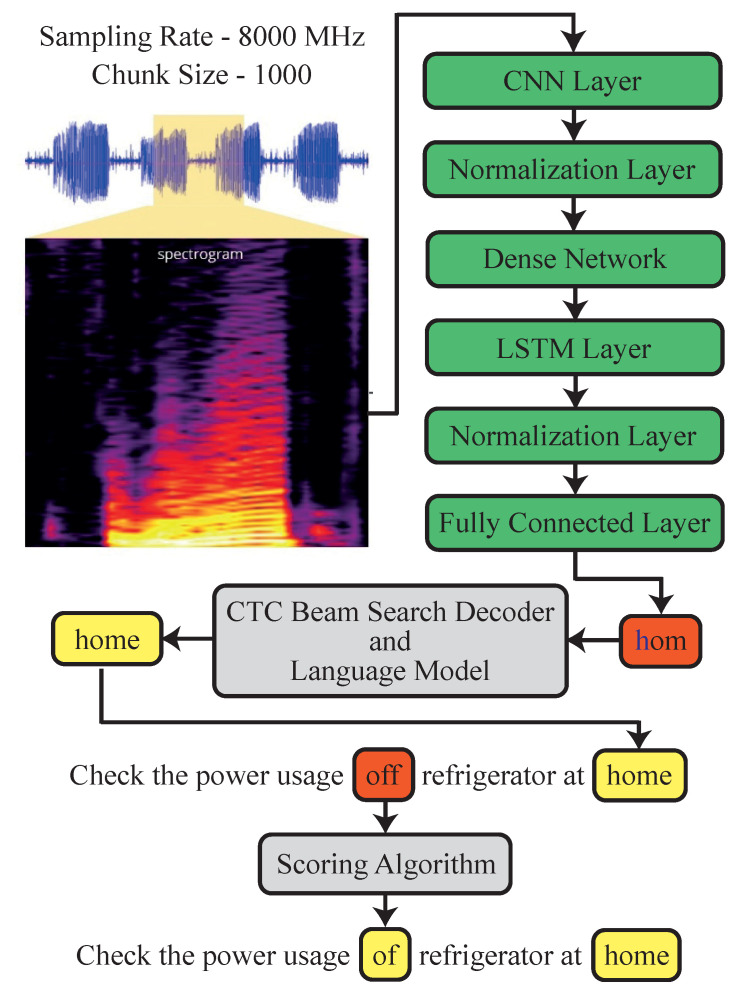
Speech to text model.

**Figure 8 sensors-23-05784-f008:**
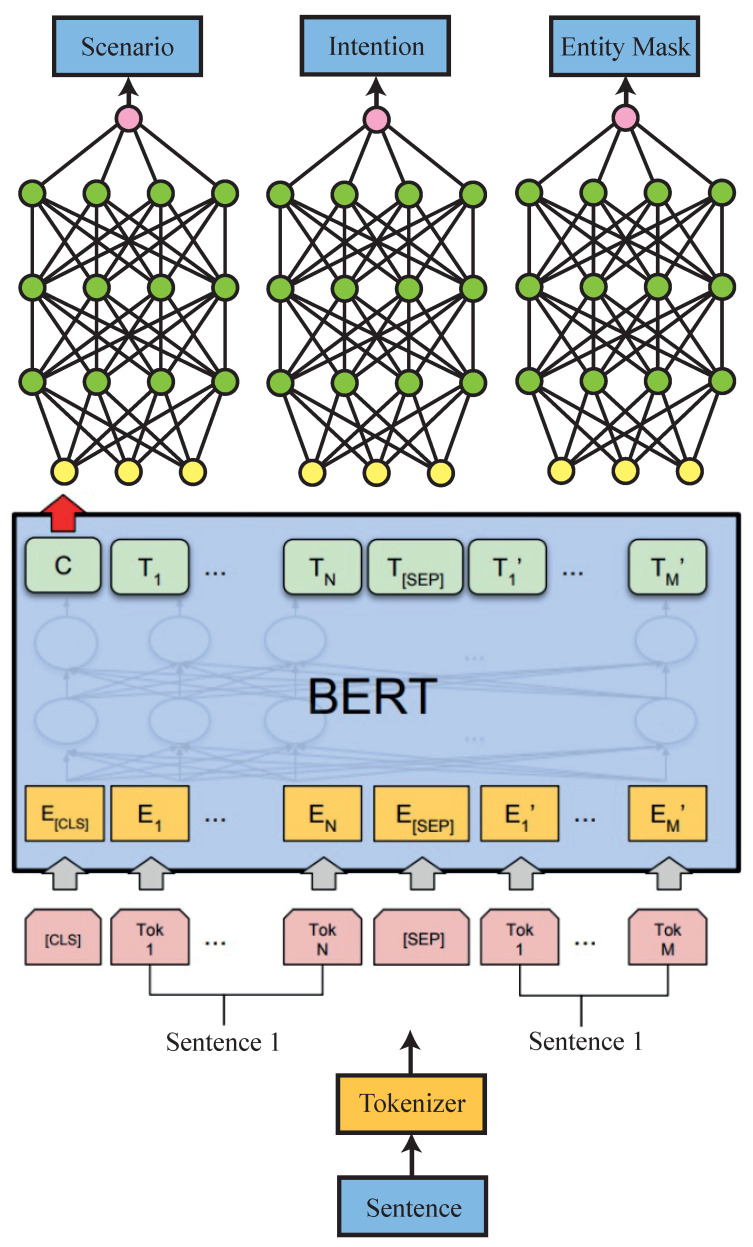
Natural language understanding model.

**Figure 9 sensors-23-05784-f009:**
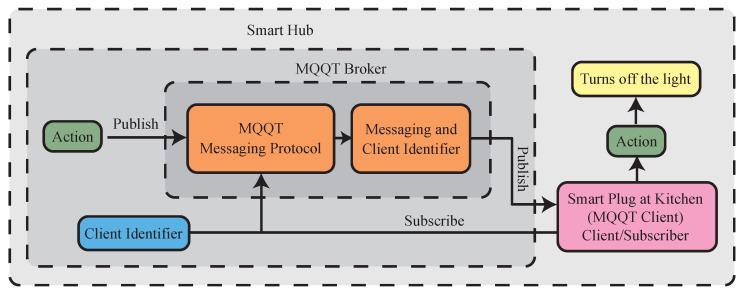
Connection between the smart hub and smart plug socket using MQQT.

**Figure 10 sensors-23-05784-f010:**
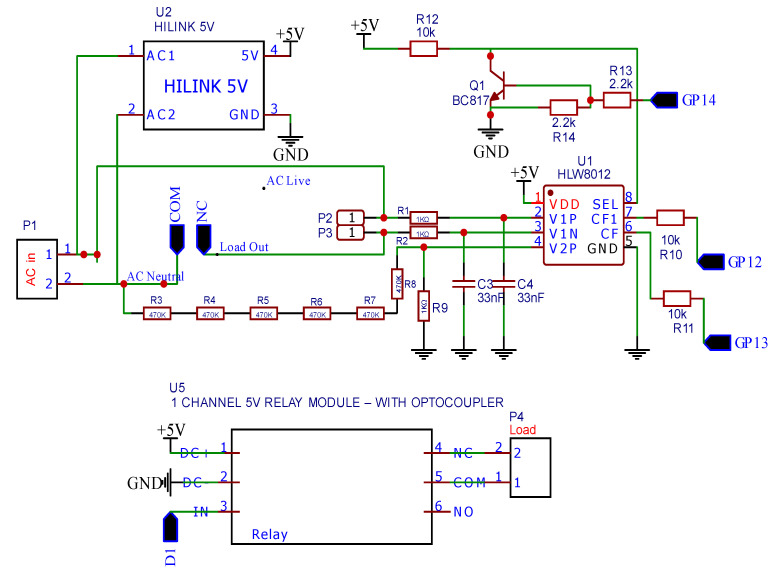
The schematic diagram of the smart plug socket.

**Figure 11 sensors-23-05784-f011:**
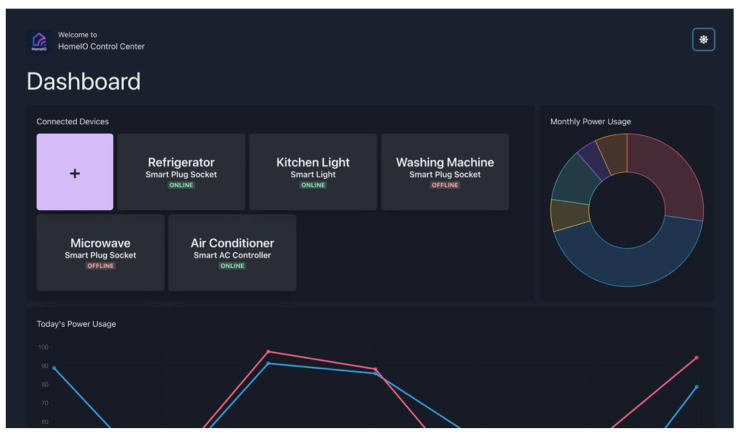
Dashboard interface.

**Figure 12 sensors-23-05784-f012:**
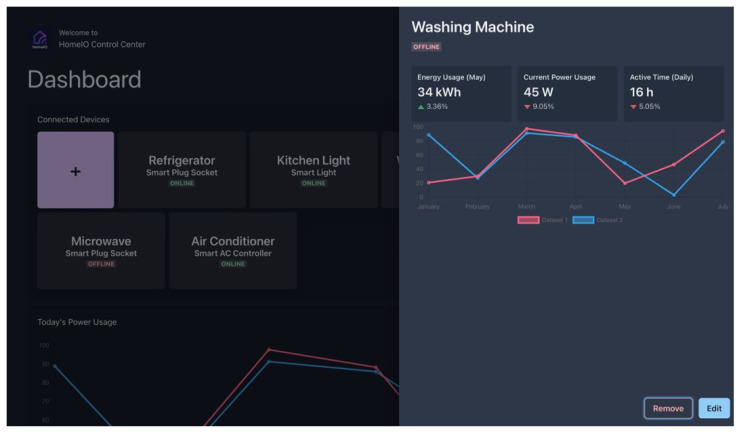
Power usage shown on the dashboard.

**Table 1 sensors-23-05784-t001:** Performance comparison of the existing STT models and our model.

Models	Model Size (MB)	No of Parameters (Million)	WER
Jasper [18]	1230	201	3.23
Wav2Letter++ [19]	2870	208	3.26
Quartz Net 15 × 5 [17]	81.1	18.9	2.96
CMU-Sphinx (HMM) [20]	70	-	11.4
Deep Speech 2 [21]	1100	47.2	6.71
Our model (Not Optimized, No LM)	55.5	4.3	8.14
Our model (traced, No LM)	18.5	4.3	6.30
Our model (traced, with LM)	180	4.3	4.61

## Data Availability

Not applicable.

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
