# Peer review of "A Secure and Smart Home Automation System with Speech Recognition and Power Measurement Capabilitiesâ€"

_sensors, 2023, doi:10.3390/s23135784_

Round 1
Reviewer 1 Report
Dear Author,
We appreciate your submission to MDPI and acknowledge the effort put into writing the paper. However, we regret to inform you that the paper falls short of publication standards due to limitations in the results presented. To enhance the chances of acceptance, we suggest that you address the following comments and recommendations:
- To provide context for your work, we recommend adding a literature review section and a review table to clarify the research gap. Your contribution to the field should be explicitly stated in the manuscript.
- Please explain in detail how your work is novel and how it advances the field. The current results are inadequate in demonstrating the effectiveness of the proposed scheme.
- We suggest comparing your results with existing techniques in the literature and providing a thorough evaluation.
- To showcase the superiority of your method, please explain how it outperforms other state-of-the-art techniques in the field.
- Additionally, providing a comparative analysis of your proposed approach compared to other existing methods would be beneficial.
- A roadmap figure can be added to the manuscript to make it easier to follow for the reader.
- The quality of the figures in the manuscript needs improvement to meet the journal standards. Please provide clearer and higher quality figures.
- Lastly, we noticed several grammatical errors in the manuscript that need to be corrected.
We hope these suggestions are helpful, and we look forward to reviewing your improved submission.
Best regards.
Reviewer 2 Report
This study suggests an offline home automation system that is not dependent on the internet or cloud services. This system protects against cyber-attacks, responds quickly, and includes capabilities such as power use tracking and optimization of linked devices.
The author contributions should be included as a separate section and in that section the author should elaborate the novelty in the proposed system compared to the existing similar systems.
The author are required to compare the results with some similar results in the literature.
Reviewer 3 Report
This paper is about home automation with speech recognition. It is extended version of the authors' previous study. The paper is well-written., but it has some restrictions:
The figures in section 2 should be placed more relevant place.
The future works should be given in conclusion.
The authors should highlight their contributions and the superiorites of their study.
The authors should give reference to their previous study in the reference list.
Round 2
Reviewer 1 Report
accepted
Reviewer 3 Report
The paper can be accepted as it is.